# Influence of the Structure on Magnetic Properties of Calcium-Phosphate Systems Doped with Iron and Vanadium Ions

**DOI:** 10.3390/ijms24087366

**Published:** 2023-04-17

**Authors:** Dania Racolta, Constantin Andronache, Maria Balasoiu, Leonard Mihaly-Cozmuta, Vadim Sikolenko, Oleg Orelovich, Andrey Rogachev, Gheorghe Borodi, Gheorghe Iepure

**Affiliations:** 1Faculty of Sciences, North University Center Baia Mare, Technical University of Cluj Napoca, 430122 Baia Mare, Romania; 2Joint Institute for Nuclear Research, Dubna 141980, Russia; 3Horia Hulubei National Institute of Physics and Nuclear Engineering, 077125 Magurele, Romania; 4R&D CSMBA, Faculty of Physics, West University of Timisoara, 300223 Timișoara, Romania; 5Karlsruhe Institute of Technology, 76131 Karlsruhe, Germany; 6REC “Functional Materials”, Immanuel Kant Baltic Federal University, Kaliningrad 236016, Russia; 7Moscow Institute of Physics and Technology, Dolgoprudniy 141701, Russia; 8National Institute for Research and Development of Isotopic and Molecular Technologies, 400293 Cluj-Napoca, Romania

**Keywords:** calcium phosphate glasses, XRD, EPR, magnetic susceptibility

## Abstract

The aim of this study was to prepare and characterize the glasses made of x(Fe_2_O_3_∙V_2_O_5_)∙(100 − x)[P_2_O_5_∙CaO] with x ranging of 0–50%. The contribution of Fe_2_O_3_ and V_2_O_5_ amount on the structure of P_2_O_5_·CaO matrix was investigated. The vitreous materials were characterized by XRD (X-ray diffraction analysis), EPR (Electron Paramagnetic Resonance) spectroscopy, and magnetic susceptibility measurements. A hyperfine structure typical for isolated V^4+^ ions was noticed to all spectra containing low amount of V_2_O_5._ The XRD spectra show the amorphous nature of samples, apart x = 50%. An overlap of the EPR spectrum of a broad line without the hyperfine structure characteristic of clustered ions was observed with increasing V_2_O_5_ content. The results of magnetic susceptibility measurements explain the antiferromagnetic or ferromagnetic interactions expressed between the iron and vanadium ions in the investigated glass.

## 1. Introduction

Studies of phosphate glasses have attracted great interest from science, engineering, and technological fields due to their valuable physical properties, which are different from silicate and borate glasses [1,2]. These properties include a lower melting temperature, high transparency in the UV domain, reduced viscosity, low glass transition temperatures, and elevated thermal expansion coefficients [1,2]. Phosphorus-based glassy materials have a disordered and partially disordered structure and by doping with different transition metal ions acquire some special electrical, optical and magnetic properties [3,4,5,6,7] compared to crystalline materials. In this way, they become useful in many fields such as electronics, optics, sealing materials, bio-glass fabrication, and also in some cases in microbiological and biomedical applications [8,9,10,11]. By adding various elements and transition metals such as iron, vanadium, and copper oxides to the phosphate glass network, the dissolution rate of the new compounds decreases while their chemical durability increases [12].

The addition of elements such as calcium, sodium, and lithium also make the compounds suitable as biomaterials [13,14,15,16,17]. Calcium phosphate materials are currently known as materials used by the human body to build bone or to automatically produce material for bone repair and regeneration. Some of them are osteoconductive and others are osteoinductive [18]. 

The forming oxide in phosphate glasses is P2O5. It has a different structure compared to other glass formers due to the existence of a terminal oxygen on each network cation. This exhibits a covalent P=O double bond and influences the additional valence electron. In fact, the P_2_O_5_ glass structure consists of a network in which three of the oxygens are bridged (P-O-P) and one is non-bridged (P=O). The basic elements of the network are PO_4_ tetrahedra linked by covalent oxygen bridges to form different phosphate anions [19]. The number of oxygen bonds present in the phosphate tetrahedra are used to define the structure of the phosphate glasses.

The properties of phosphate glasses can be modified by adding alkali and alkaline-earth oxides, such as CaO or Li_2_CO_3_, to the glass network, thereby achieving a partial structural modification [20,21]. In this way, non-bridging oxygens are created at the expense of bridging oxygens, leading to depolymerization of the phosphate network.

Ferric or ferrous oxides (e.g., Fe_2_O_3_ or FeO) confer interesting effects on the structure and properties of the phosphate-based glasses. Among the two iron ions, only Fe^3+^ shows EPR absorptions at room temperature, although both Fe^3+^ and Fe^2+^ display paramagnetic properties [22]. Although Vanadium ions incorporated in the phosphate glasses present two oxidation states, V^4+^ and V^5+^, only vanadium in its +4 oxidation state is paramagnetic. In the oxide matrix of the glass, the V^4+^ ions tend to form their specific VO^2+^ complexes. As a consequence of hopping an unpaired 3d^1^ electron from the V^4+^ site to the V^5+^ site, electrical conduction occurs. The possibility of the vanadium ion to change its oxidation state during the melting and quenching process of glass preparation was reported in the literature [23,24,25,26]. The data reveal structural changes of the units due to the formation of bonds between non-bridging oxygen and iron atoms in different coordination, while the increase in the content of vanadium ions in the samples leads to the decrease of the activation energy [23,24,25,26].

The present paper reports structural and magnetic investigations on new calcium phosphate glasses systems. The synthesis, structural, and magnetic properties of these glass systems, based on CaO and P_2_O_5_ as network formers doped with Fe and V ions, were investigated by XRD, EPR, and magnetic susceptibility measurements in a large concentration range with x varying between in the range 0 ≤ x ≤ 50%.

XRD analyzes have been employed to characterize the short-range order and electronic structures of the samples, as well as earlier for other types of glassy materials [27,28].

The EPR spectroscopy and magnetic susceptibility measurements are used to obtain complementary data regarding the influence of Fe_2_O_3_·V_2_O_5_ content on the local symmetry, and interactions between iron and vanadium ions in the P_2_O_5_·CaO glass matrix.

Investigations by the EPR technique have been reported on similar phosphate glasses to provide the most direct and accurate descriptions of the ground states and neighborhood effects on the energy levels of the paramagnetic centers, and to make it possible to determine the crystal field parameters [29,30]. 

The concentration of the 3D transition metal ions combined with the ratio of valence states and the structure of the glassy matrix influence the magnetic properties of sample. Through magnetic susceptibility measurements, the valence states of transition metal ions and the type of interactions involving them can be measured [31]. An antiferromagnetic coupling of Fe_2_O_3_-P_2_O_5_-CaO and Fe_2_O_3_-V_2_O_5_-P_2_O_5_-CaO can be considered responsible for the super-exchange interaction of the iron ions in the oxide glasses. The magnetic properties and the antiferromagnetic coupling between iron ions in different phosphate, borate, aluminosilicate, and oxide glasses have been reported previously [32,33,34,35].

The sample preparation conditions, the structure of the glass matrix, and the Fe^3+^/Fe^2+^ ratio determine the concentration range in which antiferromagnetic interactions occur. 

## 2. Results and Discussion

### 2.1. XRD Data

The X-ray powder diffraction pattern for x(Fe_2_O_3_∙V_2_O_5_)∙(100 − x)[P_2_O_5_∙CaO] glass systems are presented in Figure 1a,b. It is observed that apart from sample x = 50 mol%, all the other samples do not show characteristic diffraction peaks for crystalline phases, these being in an amorphous state. The diffraction lines for sample x = 50mol% (Figure 1b) are attributed to the crystalline phase V_3_O_5_ (PDF: 72-0524), which crystallizes in the monoclinic system having the space group Cc and the following lattice parameters: a = 9.98 Å, b = 5.03 Å, c = 9.84 Å, and β = 138.8^0^. For all other samples with 0 ≤ x < 50 mol%, it can be noted that they each have a diffraction halo characteristic for the amorphous phase, which reflects the local order. The location of this broadening halo is difficult to determine precisely, but it can be stated that the position of its maximum moves to bigger angles when the sample changes from x = 0, 1, 3, 5, 10, 20, 35 mol%.

The average distance between the atoms in the first coordination sphere R can be evaluated from the position of the halo diffraction maximum using the relation R = (5λ)/(8 sinθ). From this relationship, it was found that when Fe and V ions are introduced into the matrix containing Ca and Fe ions, the average distance between the ions increases from approximately 4 Å to 4.5 Å. 

It can be seen that the diffraction halos are a little more prominent, which means that the local order also decreases with x increasing. 

### 2.2. EPR Data

The glass systems x(Fe_2_O_3_∙V_2_O_5_)∙(100 − x)[P_2_O_5_∙CaO] were investigated by applying the EPR technique for x values in the range 0–50 mol%. The EPR spectrum of the prepared samples is presented in Figure 2.

As illustrated in Figure 2, there is a strong dependence between the absorption spectral structure and transition metal content parameters. 

At low concentrations of Fe_2_O_3_·V_2_O_5_, 0.5 ≤ x ≤ 10 mol %, the resulting spectra can be considered as an overlay of two EPR signals: (i) one with a well-resolved hyperfine structure typical of isolated V^4+^ ions; (ii) the other, with a broad line without hyperfine structure, typical for associated ions (Fe^3+^ and/or V^4+^). 

For 20 ≤ x ≤ 50 mol %, the hyperfine structure and line resolution is significantly reduced, leaving only a broad line, as a result of the increase in the number of ions associated with the Fe_2_O_3_·V_2_O_5_ content.

The g~4.3 line disappears for x ≥ 20% of transitional oxides, indicating that isolated Fe^3+^ ions form either Fe^3+^-O-Fe^3+^ bonds or Fe^3+^-O-V^4+^ interaction pairs.

The appearance of the g~2 line may be caused by the dipole-dipole interactions between Fe^3+^ ions attributed to the formation of Fe^3+^-O-Fe^3+^ bonds. These interactions lead to the formation of iron ions clusters. 

The g~2 line for x ≥ mol% content is also unresolved due to the presence of Fe^3+^ → V^4+^ electron transitions, but V^4+^ → V^5+^ transitions cannot be neglected.

The resonance line evolution with the increasing of iron and vanadium ions content in the samples can be determined using the approximate relation J = I(ΔH)2, where I represents the amplitude of the resonance line, and ΔH is the line-width.

The intensity of the resonance line, J, indicates the number of active species in resonant absorption, ΔH reflects the competition between different broadening mechanisms: dipole-dipole interactions, increasing disorder in the matrix structure, and interactions between ions with different valence states. The resonance lines centered at g~4.3 and g~2.0 in the spectra are typically for Fe^3^+ and V^4+^ ions present in the oxide glasses, their prevalence depending on x concentrations (Figure 3a,b).

The typical line-width (ΔH) increases with increasing of (Fe_2_O_3_·V_2_O_5_) content for x ≤ 10 mol% (Figure 3a) and similarly for x ≤ 1% (Figure 3b), suggesting that the dipole-dipole interactions prevails among V^4+^ ions even for clustered ions.

The decrease in the line-width with the increase in (Fe_2_O_3_·V_2_O_5_) content to more than 10 mol% (Figure 3a) and x > 1% (Figure 3b) shows that, in this composition range, the super-exchange interaction become dominant between resonance centers.

The increase in the concentration of Fe and V ions in the system can explain the J = f(x) dependence. The increase in the intensity of the resonance line (J) in the low concentration range 0–5%, (Figure 3a) and 0–10% (Figure 3b) indicates that the number of absorption centers is increasing. The decrease of J in the concentration range 5–50% (Figure 3a) and 10–50% (Figure 3b) shows the formation of iron and vanadium clusters.

### 2.3. Magnetic Susceptibility Data

The correlation between temperature and the reciprocal magnetic susceptibility for the glass samples with x varying in the range 0 < x < 50 mol% is presented in Figure 4.

The temperature correlation with the reciprocal magnetic susceptibility involves a Curie–Weiss-type behavior (χ = C/T − θ) with negative paramagnetic Curie temperature, and suggests that magnetic transition ions are isolated and/or participating in dipolar interactions in this concentration range. Therefore, we can consider that these ions are randomly distributed in the vitreous matrix, being located at distances that do not allow for the magnetic overload interaction through oxygen ions.

This shows that in all the studied concentration ranges, iron, and vanadium ions participate in different proportions depending on x, in super-exchange magnetic interactions, of antiferromagnetic type, behaving magnetically similar to other oxide glasses [36,37,38]. However, the ferric oxide concentration above which magnetic super-exchange interactions occur is lower.

In this way, one can conclude that the concentration range of transition metal ions in which these interactions occur depends on the nature of the glass matrix. Using the representation 1/χ = f(T), we calculated the molar Curie constant, C_M_, for each sample. C_M_ values increase with the increase in the content of transition metals. 

The presence of magnetic ions: V^4+^ (μ_eff_ = 1.8 µB), Fe^3+^ (μ_eff_ = 5.9 µB), Fe^2+^ (μ_eff_ = 5.1 µB) is confirmed by experimental Curie constants. Next, using the relation µeff = 2.827[Cm/2x]1/2, the magnetic moments of the samples under study were calculated.

The obtained results are for x = 10 mol%, μ_eff_ = 5.3 μB (where μB is magneton Bohr), while for the samples with higher concentration, μ_eff_ decreases to μ_eff_ = 3.67 μB. In Table 1, the obtained values of CM and μ_eff_ are given for samples with x ranging from 0–50 mol%.

## 3. Materials and Methods

Glasses of the x(Fe_2_O_3_∙V_2_O_5_)∙(100 − x)[P_2_O_5_∙CaO] system were prepared. For the systems studied, reagent grade purity substances were used for analysis. Samples were obtained by a proportional weighing of components, mixed, and melted in sintered corundum crucibles at 1523 K for a 5 min timeframe, and sudden cooling (Figure 5). Eight samples of x(Fe_2_O_3_∙V_2_O_5_) ∙ (100 − x)[CaO∙Li2O] with x = 0, 1, 3, 5, 10, 20, 35 mol% were obtained.

The structural analysis did not reveal any crystalline formations in the samples up to 50 mol% Fe_2_O_3_.

Diffraction data were obtained using a Bruker D8 Advance diffractometer using a Cu X-ray tube, the diffractometer being equipped with a germanium (1 1 1) monochromator in the incident beam and a LINXEYE position detector.

The EPR studies were conducted with a Portable Adani PS8400 spectrometer, at room temperature, in the X-ray frequency band.

For EPR investigations the samples were milled and inserted into sample tubes of the same caliber to ensure the same filling factor of the resonant cavity of the spectrometer for all samples. The mass of all samples was 100 mg. 

The determination of the magnetic susceptibility (χ) of the samples was achieved by measuring the force with which a magnetic field acts on each of these glass systems, using a Faraday balance, in the temperature range 80–300 K.

For this purpose, the sample connected to an analytical balance, placed in a quartz cup attached to the end of a quartz rod, was introduced into the magnetic field of an electromagnet. The mass of the sample was measured using the analytical balance with and without the field. In order to obtain the real magnetic susceptibility of iron ions in the studied glass compounds, due to the diamagnetism of the P_2_O_5_, CaO, and Fe_2_O_3_, corrections were considered.

## 4. Conclusions

A new system of phosphate glasses doped with transition ions such as x(Fe_2_O_3_∙V_2_O_5_)∙(100 − x)[P_2_O_5_∙CaO] was obtained and investigated in a large concentration range, i.e., 0 ≤ x ≤ 50%.

In this matrix system, P-O bonds are broken and P-O-Fe bonds are formed when iron ions are added. The gradual decrease in the number of bridging oxygen ions between Fe-O-Fe and Fe-O-V is determined by the increase in the iron ions content of the samples.

In calcium phosphate glasses, vanadium is found as isolated V^4+^ ions in C4V coordination for small transitional metal oxide content.

The XRD spectra of the glass systems x(Fe_2_O_3_∙V_2_O_5_)∙(100 − x)[P_2_O_5_∙CaO] show that, apart from the x = 50%mol sample, all the other samples do not show diffraction peaks characteristic of crystalline phases, those being in an amorphous state.

EPR measurements show that at low concentrations of Fe_2_O_3_∙V_2_O_5_, we find the presence of the two species of magnetic ions Fe^3+^ and V^4+^ in isolated positions, respectively, with the resolved hyperfine structure, the dominant interaction being the dipole-dipole one. With the increase in the concentration, the interactions of super-exchange and pair formation become dominant. The binding of isolated Fe^3+^ ions is carried out either through Fe^3+^-O-Fe^3+^ bridges, or they interact as Fe^3+^-O-V^4+^ pairs.

The magnetic measurements show that the transition ions in the glasses of the system x(Fe_2_O_3_.V_2_O_5_)·(100 − x)[P_2_O_5_·CaO] are in the valence states V^4+^, V^5+^, Fe^3+^, and Fe^2+^. For increased concentrations, the inverse of the magnetic susceptibility is determined by a Curie–Weiss-type law with negative Curie temperature. 

In the preparation of the new phosphate glasses, the addition of transition metal, such as iron and vanadium ions, at various concentrations, increases the chemical resistance of the new compound. Compared to other types of phosphate glass, we believe that the new compounds containing calcium oxide as network modifiers could become good candidates for applications in various fields such as, for example, biomedical application, and tissue engineering.

## Figures and Tables

**Figure 1 ijms-24-07366-f001:**
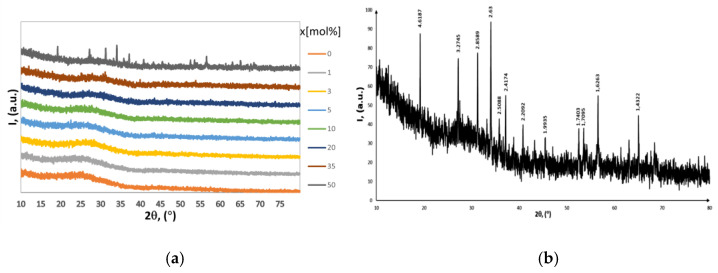
(**a**) The XRD patterns of x(Fe_2_O_3_∙V_2_O_5_)∙(100 − x)[P_2_O_5_∙CaO] glasses systems (0 ≤ x ≤ 50 mol%), (**b**) The XRD pattern of x(Fe_2_O_3_∙V_2_O_5_)∙(100 − x)[P_2_O_5_∙CaO] for x = 50%mol, normalized to maximum.

**Figure 2 ijms-24-07366-f002:**
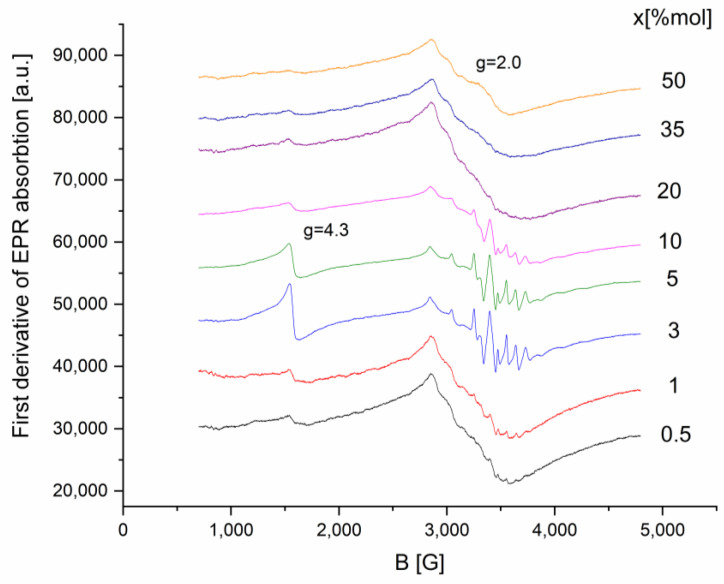
The first derivative of EPR absorption for x(Fe_2_O_3_∙V_2_O_5_)∙(100 − x)[P_2_O_5_∙CaO] glasses systems.

**Figure 3 ijms-24-07366-f003:**
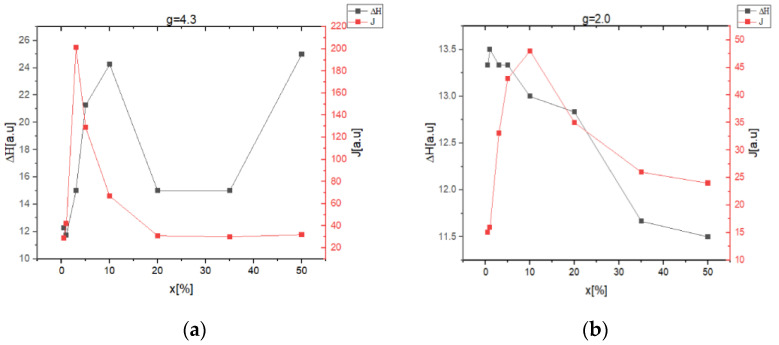
Correlation between intensity (J) of resonance line and the line-width (ΔH) for x(Fe_2_O_3_∙V_2_O_5_)∙(100 − x)[P_2_O_5_∙CaO]glass systems: (**a**) g~4.3, (**b**) g~2.0.

**Figure 4 ijms-24-07366-f004:**
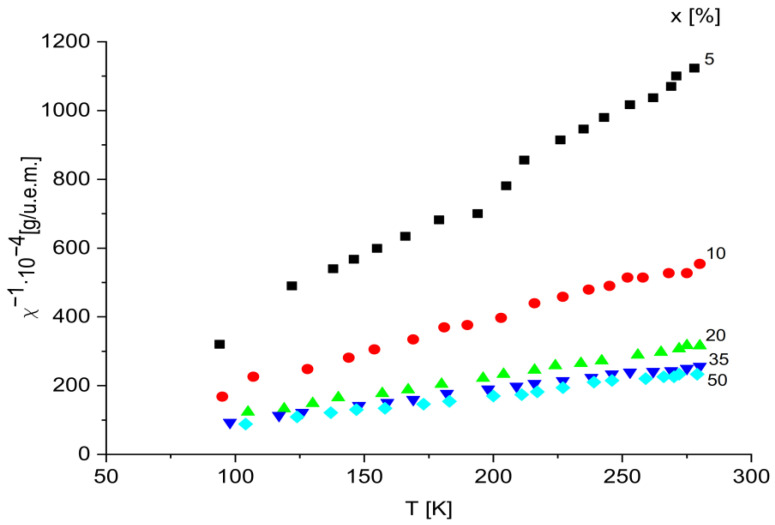
Relation between temperature and the reciprocal magnetic susceptibility for x(Fe_2_O_3_∙V_2_O_5_)∙(100 − x)[P_2_O_5_∙CaO] glasses with 0 < x < 50 mol %.

**Figure 5 ijms-24-07366-f005:**
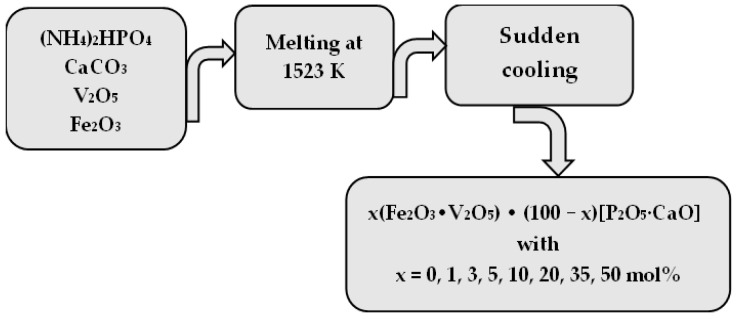
Scheme of the preparation procedure of the compounds x(Fe_2_O_3_∙V_2_O_5_)∙(100 − x)[P_2_O_5_∙CaO] with x = 0, 1, 3, 5, 10, 20, 35, 50 mol%.

**Table 1 ijms-24-07366-t001:** CM, μ_eff_ for x(Fe_2_O_3_∙V_2_O_5_)∙(100 − x)[P_2_O_5∙_CaO] with x ranging from 0–50 mol%.

x [%]	CM [u.e.m./mol]	μ_eff_ [μB]
5	0.73	5.40
10 20 35	1.41 2.54 3.10	5.3 5.03 4.20
50	3.38	3.67

## Data Availability

Not applicable.

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
