# Peer review of "Influence of the Structure on Magnetic Properties of Calcium-Phosphate Systems Doped with Iron and Vanadium Ions"

_ijms, 2023, doi:10.3390/ijms24087366_

Round 1
Reviewer 1 Report (New Reviewer)
Influence of the structure on magnetic properties of Calcium-Phosphate systems doped with Iron and Vanadium ions
Please revise and rewrite the scientific English language of the article carefully. There are many editing and writing errors.
Example:
Table 1. Table 1. CM, μeff for x(Fe2O3∙V2O5)∙(100-x)[P2O5∙CaO] with x ranging of 0-50 mol% 178
Some parts are so error-filled that make it very difficult to understand.
Example:
The magnetic susceptibility measurements (χ) can be determined by measuring the force with which a magnetic field acts on a sample of this glass system, using a Faraday-type balance in the range of temperature 80-300K The sample is placed in a quartz cup which is fixed to the end of a quartz rod with the help of which it is introduced into the magnetic field of an electromagnet
Some places have used long and inaudible sentences starting with a verb!
Example:
Was observed that the typical line width (ΔH) increases with increasing of (Fe2O3·V2O5) content for x ≤ 10 mol% (Fig.3a) and similar for x≤ 1% (Fig.3b) which suggests that the dipole-dipole interactions prevail among V4+ ions even in the case of clustered ions.
Please use the same standard punctuation. In some places have been used commas and in some places dots.
0,73
μeff = 5.3
In the materials and methods section, it is explained exactly what proportions of Fe2O3 and V2O5 are used. In addition, the melting or sintering process should be carefully explained or referenced to be repeatable. Also, Matrix Synthesis P2O5·CaO must be carefully explained or referenced.
Please use high-quality and linear Figures.
Author Response
Answer is in the attached Word-file.

Reviewer 2 Report (New Reviewer)
It is a very interesting but very brief work, can the authors include a molecular diagram of the final structure that is acquired in this new material with vanadium and iron ions.
They can argue its application at biomedical level.
Author Response
Influence of the structure on magnetic properties of Calcium-Phosphate systems doped with Iron and Vanadium ions
It is a very interesting but very brief work, can the authors include a molecular diagram of the final structure that is acquired in this new material with vanadium and iron ions.
They can argue its application at biomedical level
Thank you very much for your comment, but the purpose of this research was the synthesis method and the study of the structural and magnetic properties of these materials. In the future, we propose an in-depth study of the biomedical applications of these new materials.
Therefore, we propose the following correction for the introduction:
“Studies of phosphate glasses have attracted great interest from science, engineering, and technological fields due to their valuable physical properties, which are different from silicate and borate glasses [1,2]. These properties include a lower melting temperature, high transparency in the UV domain, reduced viscosity, low glass transition temperatures, and elevated thermal expansion coefficients [1, 2]. Phosphorus-based glassy materials have a disordered and partially disordered structure and by doping with different transition metal ions acquire some special electrical, optical and magnetic properties [3-7] compared to crystalline materials. In this way, they become useful in many fields such as electronics, optics, sealing materials, bio-glass fabrication and also in some cases in microbiological and biomedical applications [8-11]. By adding various elements and transition metals such as iron, vanadium, copper oxides to the phosphate glass network, the dissolution rate of the new compounds decreases while their chemical durability increases [12].
The addition of elements such as calcium, sodium, and lithium also make the compounds suitable as biomaterials [13-17]. Calcium phosphate materials are currently known as materials used by the human body to build bone or to automatically produce material for bone repair and regeneration. Some of them are osteoconductive and others are osteoinductive [18].
The forming oxide in phosphate glasses is P2O5. It has a different structure compared to other glass formers due to the existence of a terminal oxygen on each network cation. This exhibits a covalent P=O double bond and influences the additional valence electron. In fact, the P2O5 glass structure consists of a network in which three of the oxygens are bridged (P-O-P) and one is non-bridged (P=O). The basic elements of the network are PO4 tetrahedra linked by covalent oxygen bridges to form different phosphate anions [19]. The number of oxygen bonds present in the phosphate tetrahedra are used to define the structure of the phosphate glasses.
The properties of phosphate glasses can be modified by adding alkali and alkaline-earth oxides, such as CaO or Li2CO3, to the glass network, thereby achieving a partial structural modification [20,21]. In this way, non-bridging oxygens are created at the expense of bridging oxygens, leading to depolymerization of the phosphate network.
Ferric or ferrous oxides (e.g., Fe2O3 or FeO) confer interesting effects on the structure and properties of the phosphate-based glasses. Among the two iron ions, only Fe3+ shows EPR absorptions at room temperature, although both Fe3+ and Fe2+ display paramagnetic properties [22]. Although Vanadium ions incorporated in the phosphate glasses present two oxidation states, V4+ and V5+, only vanadium in its +4 oxidation state is paramagnetic. In the oxide matrix of the glass, the V4+ ions tend to form their specific VO2+ complexes. As a consequence of hopping an unpaired 3d1 electron from the V4+ site to the V5+ site, electrical conduction occurs. The possibility of the vanadium ion to change its oxidation state during the melting and quenching process of glass preparation was reported in the literature [23-26]. The data reveal structural changes of the units due to the formation of bonds between non-bridging oxygen and iron atoms in different coordination, while the increase in the content of vanadium ions in the samples leads to the decrease of the activation energy [23-26].
The present paper reports structural and magnetic investigations on new calcium phosphate glasses systems. The synthesis, structural and magnetic properties of these glass systems, based on CaO and P2O5 as network formers doped with Fe and V ions were investigated by XRD, EPR and magnetic susceptibility measurements in a large concentration range with x varying between in the range 0 ≤ x ≤ 50%.
XRD analyzes have been employed to characterize the short-range order and electronic structures of the samples, as well as earlier for other types of glassy materials [27,28].
The EPR spectroscopy and magnetic susceptibility measurements are used to obtain complementary data regarding the influence of Fe2O3·V2O5 content on the local symmetry, and interactions between iron and vanadium ions in the P2O5·CaO glass matrix.
Investigations by the EPR technique have been reported on similar phosphate glasses, to provide the most direct and accurate descriptions of the ground states and neighborhood effects on the energy levels of the paramagnetic centers and to make it possible to determine the crystal field parameters [29,30].
The concentration of the 3d transition metal ions combined with the ratio of valence states and the structure of the glassy matrix influence the magnetic properties of sample. Through magnetic susceptibility measurements, the valence states of transition metal ions and the type of interactions involving them can be measured [31]. An antiferromagnetic coupling of Fe2O3-P2O5-CaO and Fe2O3-V2O5-P2O5-CaO can be considered responsible for the super-exchange interaction of the iron ions in the oxide glasses. The magnetic properties and the antiferromagnetic coupling between iron ions in different phosphate, borate, aluminosilicate and oxide glasses have been reported previously [32-35].
The sample preparation conditions, the structure of the glass matrix, and the Fe3+/Fe2+ ratio determine the concentration range in which antiferromagnetic interactions occur”.
Reviewer 3 Report (New Reviewer)
The manuscript entitled “Influence of the structure on magnetic properties of Calcium- 2 Phosphate systems doped with Iron and Vanadium ions” reports the study of phosphate glasses doped with transition ions at different concentration.
The study represents an interesting application field, but it could be reconsidered after major revisions, as following suggested:
· A greater in-depth analysis is recommended in order to highlight the innovation of the study with respect the state of the art. In particular, a deepening of the introduction section in suggested by focusing the attention on specific cases study.
· In order to give to the reader a higher perception of the analysis object, a schematic representation of the synthesis method employed and the images of the developed samples could be useful.
· The results are reported in a clear way, however a comparative discussion with the results reported in literature is suggested in order to know the realistic potential properties of the glasses developed in this study.
· In both, Introduction (line 71) and Conclusion (line 229) sections, is reported the concept of biocompatibility of the calcium and phosphate-based glasses, but it is not clear how the introduction of transition ions could modify this property.
Author Response
Answer is in the attached file.

Round 2
Reviewer 1 Report (New Reviewer)
The authors seem to have answered the questions of the reviewers, and a major revision has been made. This article can now be considered for publishing.
Author Response
Thanks for review
Reviewer 2 Report (New Reviewer)
The authors miss share the last version of their work, that makes difficult the revision.
In the line 349 of the conclusion, you affirm the use biomedical application once again, but you didn`t probe in this research so I suggest change this by a suggestion.
Author Response
Thank you very much for your comment. We propose the following correction
“In the preparation of the new phosphate glasses, the addition of transition metal, such as iron and vanadium ions, at various concentrations, increases the chemical resistance of the new compound. Compared to other types of phosphate glass, we believe that the new compounds containing calcium oxide as network modifiers could become good candidates for applications in various fields such as, for example, biomedical application and tissue engineering.”
Reviewer 3 Report (New Reviewer)
The authors have accepted the suggestions and the manuscript was improved properly. It is acceptable in the current form for publication.
Author Response
Thank you very much for your review
Round 3
Reviewer 2 Report (New Reviewer)
On line 217, figure 5 is cited, but the authors forgot to include the figure caption with the title and description.
Authors have to be more careful with what they send, they must review meticulously before submitting their articles for review.
Author Response
Reviewer 2: On line 217, figure 5 is cited, but the authors forgot to include the figure caption with the title and description.
Thank you very much for your comments. We have added legend to Figure 5 and comments in the text. We have also reworded the first paragraph in the Materials and Methods section.
This manuscript is a resubmission of an earlier submission. The following is a list of the peer review reports and author responses from that submission.
Round 1
Reviewer 1 Report
The major problem with this manuscript is the lack of quantitative analysis of the X-ray edge data and the magnetic susceptibility vs. concentration data; and the inappropriate "handwaving analysis" of the EXAFS data. The conclusions drawn are not borne out by proper analyses and, moreover, appear to be contradicted by the data. the data themselves appear to be fine, which is why I did not recommend outright rejection, but serious work needs to be done to fix this paper. Additionally, some quantities are not defined, making this an arduous task for the reader.
Specific:
Lines 19, 84, and others: The EPR spectrum is incorrectly referred to as, "the EPR absorption signal," in the text. The EPR spectrum is, in fact, the first derivative of the absorption signal (ignoring artifacts derived from a finite modulation amplitude), as is correctly used in the y-axis of Figure 1. Please remove any references to "the EPR absorption signal," and replace with e.g. "the EPR signal," or "the EPR spectrum."
Line 62 and others: "XAFS" is a confusing abbreviation for X-ray Absorption Near Edge Structure and is more often used as an alternative of EXAFS; The more common, and far less confusing abbreviation for X-ray Absorption Near Edge Structure is "XANES" - please change throughout.
Line 89: "transitional metals," should be, "transition metals."
Lines 90 - 92: This sentence doesn't make any sense. "J" is undefined and, as units are not specified for "I"; the latter is perhaps understandable but renders any quantitative relationship between "J" and "I" undefined. Further, what the quantity "J" is cannot, therefore, be inferred from dimensional analysis. What the authors are trying to convey with this sentence needs to be made much, much clearer! (The reviewer notes that "J" can be inferred as being the area under the EPR absorption from the relationship quoted in lines 90 - 92 along with the y-axis labels on Figures 2 & 3. The reader should not have to search for this and the point stands that Lines 90 - 92 require much clarification in the text.)
Line 100: Please make it clear that it is the RESOLUTION of the hyperfine that is diminished, not the hyperfine coupling constant.
Line 106: What is meant by "scattering?"
Line 120 and Figure 4. The edge position is usually taken as the maximum in the first derivative of the XAS spectrum in the edge region. The data of Fig 4 would appear to suggest that the edge positions of 0.20 % Fe and 0.50 % Fe are very similar, while the intermediate composition of 0.35 % Fe has a higher-energy edge position. Therefore the statement of line 120, and particularly the use of the word, "unambiguous," is not warranted by the data. A complication may be that the edge structure is complicated by pre-edge non-ionizing electronic transitions. Showing the first derivative spectra would provide a basis from which to discuss the edges in more detail. At present, the data appear to contradict the claims of lines 118 - 124.
Lines 131 - 133. The claim that the 1.5 A peak in the FT spectrum is due to, "reorientation of Fe-O octahedra," is highly specious. A serious flaw with this manuscript is the lack of quantitative analysis of the EXAFS data. The common procedure is to fit the edge-normalized background-subtracted EXAFS data in k-space (reciprocal Angstroms) - this is the ONLY way to get meaningful numbers for ligand atoms via EXAFS because the FT spectrum contains only the absorption information and none of the phase information. Without this type of analysis, the FTs of the EXAFS data are quantitatively meaningless.
Lines 145 - 153. To this reviewer, there is no qualitative difference in the magnetic susceptibility data; ALL of the magnetic susceptibilities appear to vanish at a negative Curie temperature. However, the main point as far as reviewing this manuscript is that NO FITS TO THE DATA ARE PRESENTED that show the extrapolation to the temperature at which the M.S. = 0, and there are no statistics to show the confidence with which those extrapolated temperatures could be taken. Again, this is a serious flaw and invalidates the interpretation of the magnetic susceptibility data.
Figure 7. "S5" and "S6" are not defined, so the figure is completely without value.
Author Response
Please find in attached file the answer to review

Reviewer 2 Report
Major revision of the manuscript as per comments is required
Author Response

(The authors gave the same response as above.)

Round 2
Reviewer 1 Report
There is no reason why the ORIGINAL X-ray absorption data cannot be properly fitted.
Reviewer 2 Report
The manuscript is acceptable after this major revision
